# Ultra-Broadband, Polarization-Irrelevant Near-Perfect Absorber Based on Composite Structure

**DOI:** 10.3390/mi13020267

**Published:** 2022-02-06

**Authors:** Yanlong Meng, Jinghao Wu, Simeng Liu, Yi Li, Bo Hu, Shangzhong Jin

**Affiliations:** 1The Postdoctoral Center of the Department of Electronic Engineering, Fudan University, Shanghai 200433, China; mengyanlong@yeah.net; 2State Key Laboratory of Applied Optics, Changchun Institute of Optics, Fine Mechanics and Physics, Chinese Academy of Sciences, Changchun 130033, China; 3College of Optical and Electronic Technology, China Jiliang University, Hangzhou 310018, China; wwjh1104@163.com (J.W.); P1904085222@cjlu.edu.cn (S.L.); yli@cjlu.edu.cn (Y.L.); szjin@cjlu.edu.cn (S.J.)

**Keywords:** absorption, ultra-broadband, multilayer, nanostructure

## Abstract

This paper proposes a near-perfect absorption device based on a cross-shaped titanium nanostructure and a multilayered structure. The multilayered bottom structure consists of alternately SiO_2_ and Ti. The whole device is put on a TiN substrate. The coupling between cross-shaped titanium nanostructures, and that between the cross-shaped titanium nanostructure and bottom multilayer, can further enhance the absorption at some wavelength where most of the energy is reflected or passes through in the device with a single structure. According to the simulation results, the device presents a nearly perfect absorption in a wavelength range from 300 nm to 2000 nm. The average absorptance in the wavelength range from 500 nm to 1400 nm exceeds 96%. This paper also provides a new idea for realizing perfect absorption, which is extensively used in sensing, controllable thermal emission, solar energy harvesting solar thermo-photovoltaic devices, and optoelectronic metrology.

## 1. Introduction

As a kind of green energy, solar energy plays an important role in replacing traditional fossil fuels [1]. Its vast application prospects attract a large number of researchers to explore a truly highly-effective utilization manner for such energy. Commonly, solar energy is applied in two ways: solar thermal (ST) and photovoltaic (PV) devices. Regardless of the energy conversion mechanism, all these devices inevitably need to absorb the solar light incident into their surface as much as possible. Therefore, improving the absorption ability of devices utilized for solar energy conversion should be considered first. In order to realize a perfect absorption for the solar spectrum, many ways are proposed and demonstrated. However, the wavelength range concerned in most of the research is restricted to the visible range. It is a challenge to a material that consistently possesses a strong absorption coefficient in the entire solar spectrum range. In recent years, many nanostructures have been proposed in various application fields, such as structural color, perfect absorber, filters, and hyper-spectral imaging [2,3,4,5]. Herein, such a type of structure was also investigated intensively in solar energy devices, especially in solar thermal devices [6,7,8,9,10].

For solar thermal systems, achieving efficient sunlight absorption in broadband is one of the critical points for improving the conversion efficiency of the whole system. As a solar heat system, the sunlight absorber needs to be perfectly absorbed in the broadband and be insensitive to polarized light. In order to achieve such an aim, many schemes were considered, such as multiple resonances, impedance matching, and lattice scattering [11,12]. The perfect broadband absorbers realized based on those schemes were achieved by coupling the top structure of some specific patterns with the dielectric layer, and most of the materials used in the metal layer are precious metals, such as gold and silver [13,14,15,16]. But most noble metals have a relatively narrow range for spectral absorption and using noble metals to make a perfect broadband absorber is also relatively costly. Therefore, the material selection of the metal layer still needs to be optimized. In consideration of the application purpose, it is necessary to have strong absorption in a broad spectral range for those absorptive materials utilized in devices. However, its stability at high temperatures should also be considered. Therefore, some refractory metallic materials become excellent candidates, such as titanium. It has been reported that an average absorption of more than 95% in the wavelength range of 400 nm-900 nm was achieved by using titanium [17,18,19]. As a refractory material, the melting point of metal titanium is as high as 1668 °C. Because of its excellent optical loss characteristics, titanium shows a higher light absorption efficiency in a fairly broad wavelength range. The extraordinary properties of titanium make it possible to realize a perfect absorber. Therefore, extensive research on titanium metal and its chemical compounds was carried out [6,20,21,22,23]. Yu used TiO-TiN materials to make solar absorber entities, which can reach an average absorption efficiency of more than 90% in the wavelength range of 1264 nm [21]. Li et al. reported a cross-shaped TiN structure that could reach an absorptance of more than 90% at the wavelength of 1182 nm [20].

This paper proposes an ultra-broadband near-perfect absorber based on the refractory metallic material Ti and its chemical compound TiN. Since it has a low relative dielectric constant in the optical range and the melting point of SiO_2_ is also very high, it is suitable for use in the structure. The material used for the substrate is TiN, which can guarantee the full absorption of light. The material enables the designed absorber to produce a strong surface plasma resonance effect [24,25], which improves the light absorption efficiency of the structure and increases the band range of light absorption. The optimized structure can have an average absorption rate of more than 92% in the wavelength range of 300~2000 nm and up to 96% in the 500~1400 nm wavelength range. In addition, the structure is insensitive to the incident angle of light.

## 2. Simulation Model

The absorber structure proposed in this paper is shown in Figure 1. As shown in Figure 1a, a cross-shaped nanopillar array was set on the top of the multilayer structure, of which the material was titanium. Two types of cross-shaped nanopillars with different widths were designed in one period (see Figure 1a). However, the two kinds of cross-shaped nanopillars have the same height, length, and arm width. To depict and discuss our results clearly, we defined the period of the array as P. The length, arm width, and height of the two kinds of cross-shaped nanopillars were defined as M, W, and H1, respectively. The widths of the two crosses were defined as W3, and W4, respectively. The bottom multilayer was formed by stacking Ti and SiO_2_ layer alternately_._ A thick titanium nitride substrate was used in the designed structure to ensure the complete absorption of light. All the specific parameters of the structure proposed in this paper are listed in Table 1.

The research method used in this paper is the time-domain finite-difference (FDTD) analysis [15,19]. The simulating environment is air. Accordingly, the index of background is set to 1. We set the boundary conditions in the X and Y directions of the structure to periodic boundary conditions. The boundary condition in the Z direction was a perfect match layer. The light source used in the simulation was a plane wave, of which the spectrum range was from 300 nm to 2000 nm. The incident light propagated along the Z backward. The optical constants of materials used in the simulation are plotted in Figure 2, which are quoted from PALIK’s database, then we set up a power monitor directly above the absorber to get the data of reflected electric power. Since there is no light passing through the absorber, absorptance *A* can be obtained by the formula A=1−R. Three electric field monitors were put in different positions to get the distribution of electric and magnetic fields to facilitate subsequent analysis.

## 3. Results and Discussion

Figure 3 shows reflectance and absorptance curves of different absorbers, including composite structure, a multilayer structure without top nanopillars, and a pure nanopillar array. It can be seen from the figure that the absorptance of the absorber without a cross-shaped structure presents an obvious reduction, in comparison with that of the absorber with a composite structure, in the range from 300 nm to 800 nm. Absorption of the absorber without the cross-shaped structure is also weakened in the range of 1000–1500 nm. Though the absorber without a top cross-shaped structure shows a high average absorptance, introducing a cross-shaped structure on top of the multilayer structure certainly helps the bottom multilayer to improve the total absorption. Then, the overall absorption increases from 0.88 to 0.96. It is apparent in Figure 2 that the pure cross-shaped nanopillar array shows a strong absorption right in the range from 300 nm to 600 nm, which is right complementary to that of the bottom multilayer. As a result of those complement absorptive characteristics, the absorption of composite structures displays a near-perfect absorption effect in broadband. The absorptance in the full wavelength range exceeds 0.92 and even exceeds 0.96 from 500 nm to 2000 nm. It also can be seen from the figure that there are two valleys in the absorptance curve at 455 nm and 868 nm, corresponding to the absorptance of 0.92 and 0.97, respectively. The maximum absorption is close to 1.0 at 620 nm. The results show that the structure can achieve a near-perfect absorption effect in broadband. Since there is no light passing through the absorber with the composite structure, only absorption and reflection occur as the light incident on the absorber. The near-perfect absorption also means there is almost no reflection in the absorber (See Figure 2). To further illustrate the reason for obtaining high absorption in the broadband based on the composite structure, we compared electric field distribution at different cross-sections of the composite structure, as shown in Figure 4. 

To study the physical mechanism of the enhancement in light absorption by introducing the cross structure, we simulated the distribution of electric fields at the spectral valleys and peaks. The calculated electric field distribution in different cross-sections was obtained by three different field monitors, which is shown in the right-down corner of Figure 4. Figure 4a,d,g,j depict the electric field distributions in the cross-section of the cross, which are obtained from monitor 1. Figure 4b,e,h,k shows the electric field distributions in a perpendicular plane between two crosses, which are obtained from monitor 2. Figure 4c,f,i,l present the electric field distributions in the cross-section of the cross-shaped structure, which are obtained from monitor 3. From Figure 4a–c, it can be seen that the electric field around the top cross-shaped structure shows a high intensity at 455 nm. However, the electric field in the cross structures’ gap, i.e., the electric field recorded by monitor 2, is weakened due to a weak plasma resonance. The weak plasma resonance can’t sustain a strong absorption, leading to energy leakage. Figure 4d–f show the electric field distribution at 620 nm, where the strongest absorption occurs. It can be seen from the graph that strong electric fields exist not only around the cross structure but also in the gap between the two cross structures. The distribution of the electric field indicates that there is a strong plasma resonance coupling at this wavelength. The strong coupling leads to strong energy absorption. Lastly, the absorptance at this wavelength is improved. Figure 4g–i shows an electric field distribution at the wavelength of 868 nm. It can be seen from these figures that the electric field distribution is still around the cross structure, but the electric field in the middle gap is weakening. However, it is still higher than the intensity of the electric field at the wavelength of 455 nm. As a result, the absorption of the device at this wavelength still maintains high efficiency. Figure 4j–l are the electric field distributions at a wavelength of 1210 nm. It is evident that the electric field is mainly distributed at the four ends and at the top of the cross structure. In addition, there is a certain electric field distribution at the cross structure and the underlying intersection. Accordingly, the plasma effect is enhanced. Then, strong light absorption occurs. According to the evolution of electric fields at various wavelengths and positions, it is found that the electric field is mainly limited around the cross structure. The results also reveal that the plasma resonance strongly affects absorption enhancement in the composite structure.

To further illustrate the relationship between light absorption and the parameters of each part in the structure, we investigated the evolution of absorption as the parameters of the structure change. Firstly, the change of absorption of the absorber is investigated as the number of titanium layers changes. Figure 5 shows the absorptive curves of the absorbers with different numbers of layers in the bottom multilayer, as the thickness of each titanium layer is fixed at 8 nm. The results manifest that there is no need to realize a high absorption by continuously increasing the number of titanium layers. The best scheme is two layers of titanium for this structure. As the number of titanium layers exceeds two, the overall light absorption efficiency begins to degrade. When the number of titanium layers increases to two, the minimum absorptance of the device also increases from 0.80 to 0.92. However, if we increase the number of titanium layers in the absorber further, it is found that the absorptance tends to decline. The results reveal that, when the underlying structure is coupled with the cross structure, it is unnecessary to improve the overall absorption by increasing the number of titanium layers. Therefore, the absorption reaches the strongest because of the strong plasma resonant coupling when the number of metal layers is two.

Since the number of bottom titanium layers is determined, the influence of the metal thickness on the absorption is studied further. Figure 6 shows the absorptive curves of absorbers with various thicknesses of titanium. It can be seen from the diagram that the overall absorptance increases at the beginning. As the thickness of the titanium layer increases, the absorptance in the short-wavelength range tends to decline. As shown in Figure 6, there is a notable decrease in absorption in the range from 400 nm to 500 nm as the thickness of the Ti layer increases to 10 nm. The trend is similar to that of the change as the number of Ti layers changes. Therefore, for the metal layer, we choose a thickness of 8 nm as the thickness of this structure.

In addition to studying the underlying metal layer, we also discussed the effect of silica in the bottom multilayer. As shown in Figure 7, we changed the thickness of the silica layer above the titanium nitride from 40 nm to 100 nm. It can be seen from the figure that the absorbing efficiency of light increases first and begins to decline as the thickness of the silica increases to 60 nm in the short-wavelength range. When the thickness of silica exceeds 80 nm, the short-wavelength light absorption exhibits a significant degradation. Accordingly, the thickness of 60 nm is chosen as the optimal silica thickness.

Since this absorber is composed of a nano-cross array and a multilayer plane structure, the effect of the top nano-cross array was also investigated. The change of absorptance curves is shown in Figure 8 as the arm width of the two types of the nano-cross in one period are changed to 3 nm, 5 nm, 7 nm, and 9 nm. It can be seen from the figure that as the width of the cross structure increases from 3 nm to 9 nm, the absorptance in the range from 400 nm to 700 nm ascends while it declines in the longer wavelength range.

Figure 9 shows the absorptance curves of the designed absorber as the height of nano-cross changes to 100 nm, 150 nm, 200 nm, 250 nm, 300 nm, respectively. It can be seen from the diagram that the overall absorption efficiency presents a notable increasing trend in the range from 500 nm to 900 nm, though there is a slight decline in the range from 900 nm to 1200 nm. As the height is 250 nm, the average absorption efficiency can reach 0.96. The absorption efficiency even reached 0.97 when the height is as high as 300 nm. It is very clear that the increment of height results in an improvement in the absorption efficiency. The increase in absorption in the short-wavelength range is attributed to the enhancement of the surface plasma effect, while the dominant coupling in the long-wavelength range is the coupling in the bottom multilayer. Finally, it is more suitable for practical application when the height is 250 nm in consideration of other factors, such as processing complexity and manufacturing costs.

In order to manifest the advantages of using titanium, the total absorption of the composite absorber as the metallic material changes is plotted in Figure 10. It is clear that the worst absorption occurs as the metallic material is changed to Au and Ag, though the two materials possess a strong plasmon effect. In the wavelength range from 500 nm to 2000 nm, there is an entire reduction in absorption as the metallic material changes. The reason for such a reduction of absorption lies in the weaker extinction coefficients of the noble metals and the mismatch of the refractive index. It should be noted that there are two dramatic dips both in the absorptance curves of Au and Ag. This means strong reflections occur at these wavelengths, which are typically derived from the mismatch of the refractive index. 

Another concerning characteristic of solar absorbers for practical application is the response of the absorber to the polarization and the incident angle. Figure 11 presents the evolution of absorptance as the polarization angle and the angle of incidence change. When the angle of incidence increases from 0° to 60°, there are almost no changes in the absorptance, except in the short-wavelength range. As the polarization angle changes from 0° to 60°, the overall absorber can maintain a good absorption efficiency in the full range of 300~2000 nm. The results show that the structure of the cross and the underlying plane coupling can have a better absorptance on the broader range and present an angle-insensitivity and polarization-irrelevant.

To verify the final effect of the absorber, the distribution of absorbed solar spectral power density is calculated based on the standard solar radiation spectrum AM 1.5. As shown in Figure 12a, the black spectrum is the standard solar radiation spectrum of AM 1.5, and the red line represents the absorption spectrum of the absorber. It is straightforward that the absorber maintains strong absorption characteristics in the full spectrum range. The absorber presents a perfect absorption, especially in the range of which the wavelength is longer than 1200 nm. Figure 12b shows the total absorbed power and the energy missed. The red area stands for the absorbed energy, and the orange part is the unabsorbed energy. It can be seen that there is only little energy in the short-wavelength range missed by the absorber. Therefore, it can be seen that the resonance absorption of the solar energy in the 300~2000 nm band range has a certain advantage. The proposed absorber can play a good role in energy absorption and the solar thermal photovoltaic system.

## 4. Conclusions

In summary, an ultra-broadband near-perfect absorber based on a composite structure is investigated numerically. The coupling between the nano-cross array provides an extra absorption, which helps to enhance the total absorption of the proposed absorber. Finally, the proposed composite absorber achieves an average absorptance of more than 95% in an ultra-broad range from 300 nm to 2000 nm. Based on such structure, continuously increasing the thickness and layers’ number of titanium in the bottom multilayer structure has no remarkable benefits for enhancing the absorption of the absorber. Choosing a proper coupling manner and structures plays an important role in realizing an ultra-broadband near-perfect absorber. The proposed absorber shows a polarization-irrelevant and angle-insensitive characteristic that is very helpful for practical application. Finally, because the materials used in the absorber are titanium, titanium nitride, and silica, this ensures that the structure can be used at higher temperatures, and the material cost is low. The absorber can play an important role in the photovoltaic collection, solar photovoltaic systems, thermal electronic devices, and other fields relating to light absorption.

## Figures and Tables

**Figure 1 micromachines-13-00267-f001:**
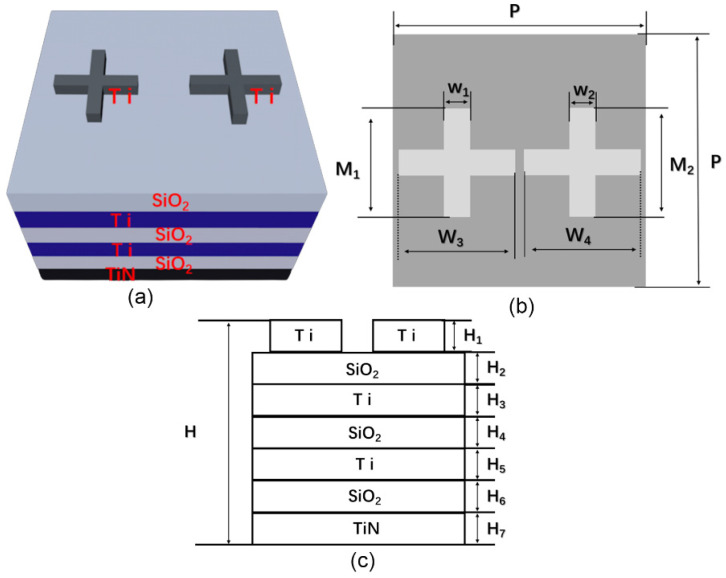
(**a**) Schematic diagram of the ultra-broadband absorber structure (**b**) Top view of the absorber. (**c**) Lateral view of the absorber.

**Figure 2 micromachines-13-00267-f002:**
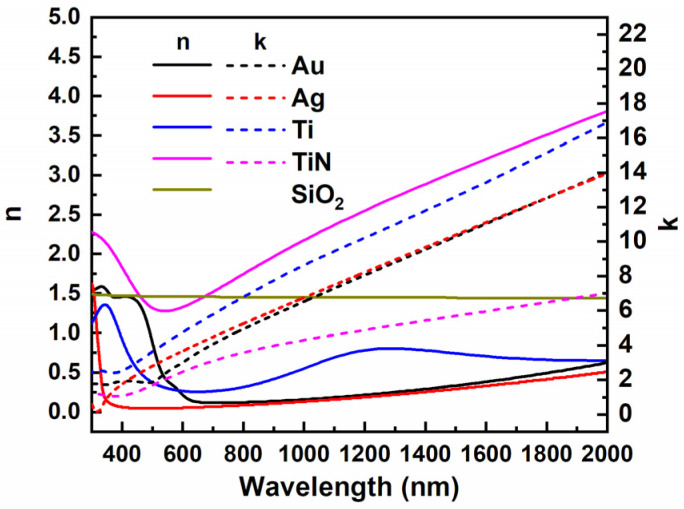
The refractive index and extinction coefficient of materials used in the simulation.

**Figure 3 micromachines-13-00267-f003:**
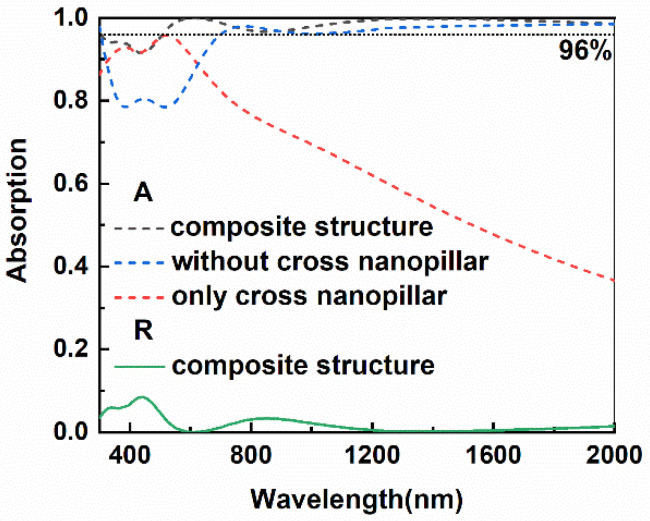
The absorptance and reflectance curves of the absorbers with different structures.

**Figure 4 micromachines-13-00267-f004:**
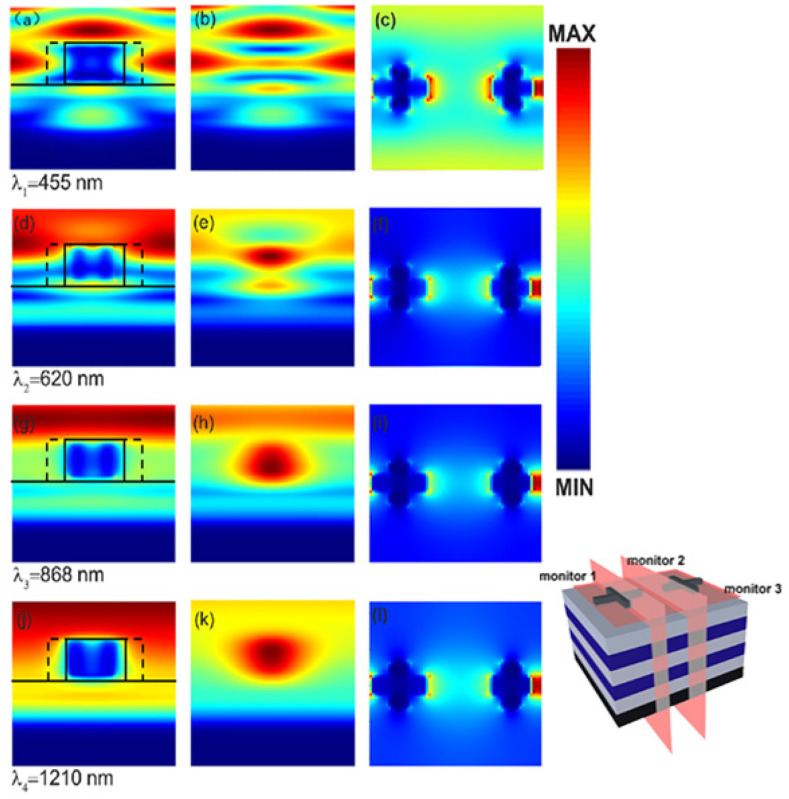
The electric field distribution at different positions as the light with various wavelengths, λ_1_ = 455 nm (**a**–**c**), λ_2_ = 620 nm (**d**–**f**), λ_3_ = 868 nm (**g**–**i**), and λ_4_ = 1210 nm (**j**–**l**), incident onto the absorber. The electric field distributions in the first column are obtained from monitor 1. The data in the second column are obtained from monitor 2. The electric field distributions in the third column are obtained from monitor 3.

**Figure 5 micromachines-13-00267-f005:**
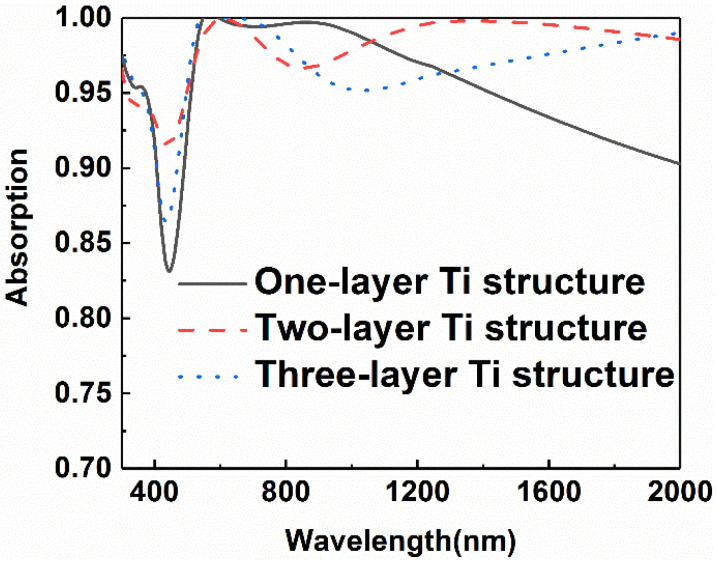
Comparison of absorption curves of the composite absorber with different titanium layer numbers in the bottom multilayer.

**Figure 6 micromachines-13-00267-f006:**
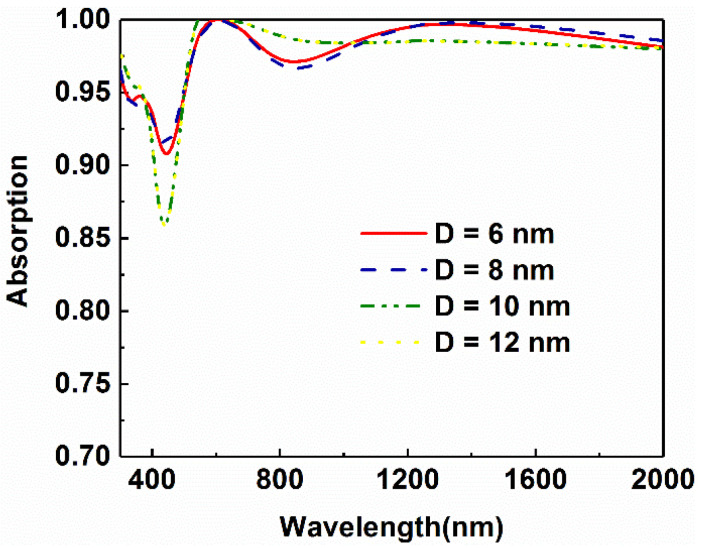
Comparison of light absorption of the absorbers with different Ti thicknesses in the bottom multilayer.

**Figure 7 micromachines-13-00267-f007:**
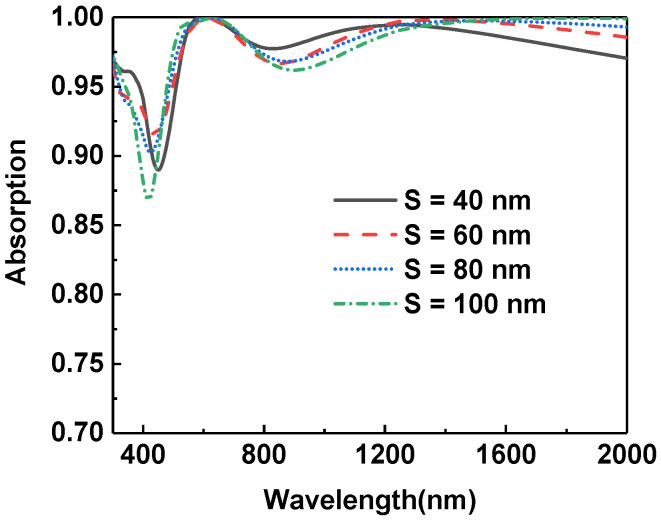
Comparison of light absorption of the absorbers with different SiO_2_ thicknesses in the bottom multilayer.

**Figure 8 micromachines-13-00267-f008:**
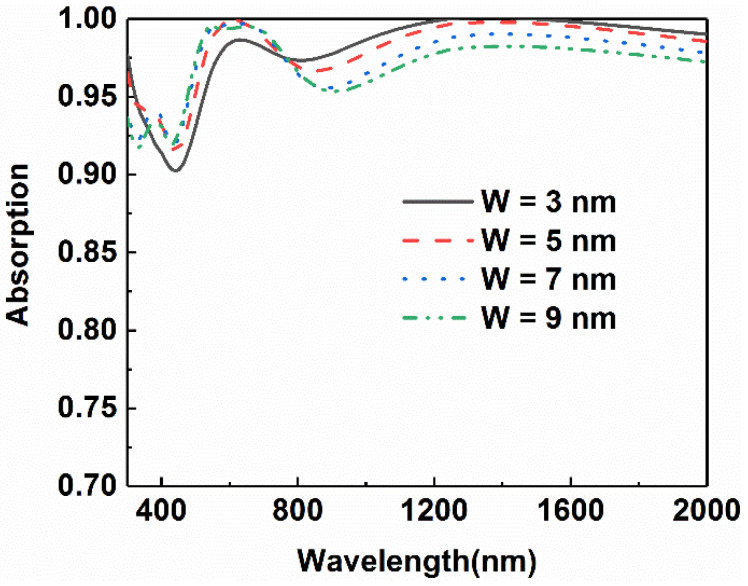
Comparison of light absorption of the absorbers with different arm widths of the cross.

**Figure 9 micromachines-13-00267-f009:**
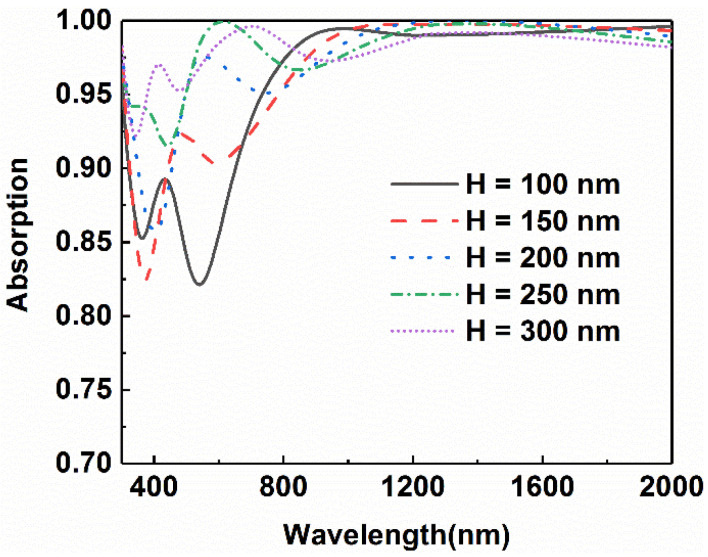
Comparison of light absorption of absorbers with different heights of the cross.

**Figure 10 micromachines-13-00267-f010:**
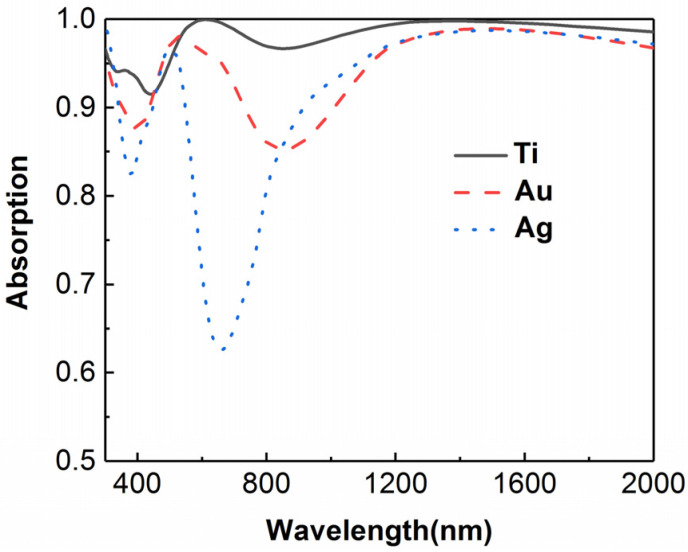
Comparison of light absorption as the metallic material changes.

**Figure 11 micromachines-13-00267-f011:**
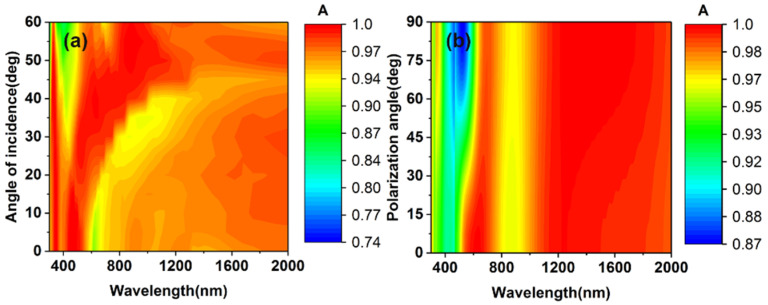
(**a**) The evolution of absorbers’ absorption as the incident angle increases from 0° to 60°. (**b**) The evolution of absorbers’ absorption as the polarization angle increases from 0° to 90°.

**Figure 12 micromachines-13-00267-f012:**
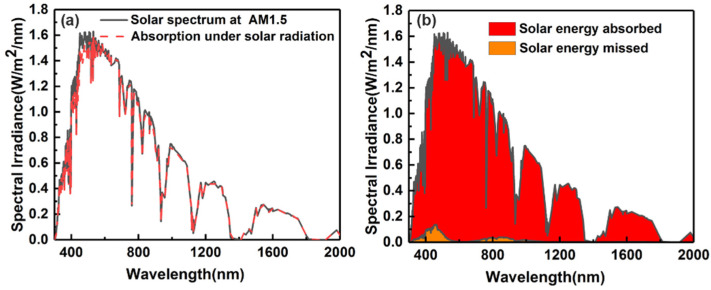
(**a**) Comparison of the AM 1.5 standard spectrum of solar radiation and the absorbed spectrum. (**b**) The energy absorbed and lost by the absorber under AM 1.5.

**Table 1 micromachines-13-00267-t001:** The structural parameters used in the simulation.

Layer	Structural Parameter			
Cross-Shaped Ti Nanopillars	H_1_ = 250 nm	M = 120 nm	W = 50 nm	W_3_ = 150 nm/W_4_ = 110 nm
SiO_2_	H_2_ = 130 nm			
Ti	H_3_ = 8 nm			
SiO_2_	H_4_ = 135 nm			
Ti	H_5_ = 8 nm			
SiO_2_	H_6_ = 60 nm			
TiN	H_7_ = 300 nm			
Period	P = 300 nm			

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
