# Peer review of "Ultra-Broadband, Polarization-Irrelevant Near-Perfect Absorber Based on Composite Structure"

_micromachines, 2022, doi:10.3390/mi13020267_

Round 1

Reviewer 1 Report

In the paper, the authors demonstrate a composite structure based near perfect absorber, which shows dramatic light absorption in the range from 300 nm to 2000 nm. This study provides a systematical discussion in terms of operating mechanisms and potential application of solar energy harvesting. Therefore, the referee suggests the publication of this manuscript after the revisions below:

  1. Please depict whether the environment of the simulation model is air or vacuum, since the environmental refractive index could affect the absorption characteristics.
  2. Since polarization-irrelevant absorber has been expected, why not design the Ti structure at the top panel as a non-oriented structure, such as spherical structure and disk arrays?
  3. There is a slight dip of absorptance at the wavelength of ~455 nm. Can the absorption capability be improved further at this wavelength, if the Ti structure be changed to plasmonic noble metals Ag?
  4. Please unify the format and refine the Figures, and add some theoretical equations to strengthen the simulation results further.

Author Response

The authors appreciate the reviewer's kind comments. All replies to the comments are listed in the attachment. Please see the attachment. 

Reviewer 2 Report

This paper has proposed an ultra-broadband near-perfect absorber based on refractory metallic material Ti and its chemical compound TiN. The optimized structure was argued to give an average absorption rate of more than 92% in the wavelength range of 300 nm 72 ~ 2000 nm and up to 96% in the 500 nm~ 1400 nm wavelength range, in which, the chemical system was shown to be insensitive to the incident angle of light. While this work may be acceptable for possible publication in a journal, I ask the authors to revise the paper properly.

  • The simulation model is not correctly described. There is no reference to the finite difference method used for the study, as written by the authors “The research method used in this paper is the time-domain finite-difference (FDTD) analysis..” According, the details of simulation must be described.
  • The quality of picture in Figure 4 is too poor. It is colored but cannot be understood.
  • The size of all other pictures should be enhanced.
  • Figure 11 needs significant elaboration. It is weakly presented.
  • A = 1 – R is written in the paper, yet Fig. 2 labels transmittance “T”. Why?
  • Figs. 2 and 3 are nearly overlapped.

Author Response

(The authors gave the same response as above.)

Reviewer 3 Report

In this manuscript, the authors investigated numerically a near-perfect absorber based on a multilayered structure, on which a cross-shaped titanium nanostructure was mounted. The simulation results may give insight into the light absorption behavior in a wavelength range from 300nm to 2000 nm, however, it should be rejected at present status. The reasons are as follows for reference:

  1. The simulation details are not adequately supplied, such as the optical constants of materials, necessary information on FDTD calculation, etc. A table of the parameters list is suggested.
  2. What’s the reason for the choice of a cross shaped Ti? Can other pattern of metal be available for the enhanced absorber?
  3. Discussion about the effects of different materials, cross-shaped titanium nanostructure, coupling between layers, and the polarizations mentioned on the performance, are always necessary.
  4. Based on a simplex simulation without care about materials’ characters and the coupling interactions, the results obtained in this research will be lack of reliability and significance.
  5. Some spelling mistakes need correction.

Author Response

(The authors gave the same response as above.)

Round 2

Reviewer 2 Report

Authors have considered my comments, and revised their paper as suggested. Whereas the language used to prepare the ms is not smooth, this interesting work may be considered for publication.

Reviewer 3 Report

Authors have carefully addressed all issues raised by the reviewer, it becomes acceptable now.